# The Potential Beneficial Effects of Resveratrol on Cardiovascular Complications in Marfan Syndrome Patients–Insights from Rodent-Based Animal Studies

**DOI:** 10.3390/ijms20051122

**Published:** 2019-03-05

**Authors:** Mitzi M. van Andel, Maarten Groenink, Aeilko H. Zwinderman, Barbara J.M. Mulder, Vivian de Waard

**Affiliations:** 1Department of Cardiology, Amsterdam UMC, Academic Medical Center, Amsterdam Cardiovascular Sciences, University of Amsterdam, 1105 AZ Amsterdam, The Netherlands; m.m.vanandel@amc.uva.nl (M.M.v.A.); m.groenink@amc.uva.nl (M.G.); b.j.mulder@amc.uva.nl (B.J.M.M.); 2Department of Radiology, Amsterdam UMC, Academic Medical Center, University of Amsterdam, 1105 AZ Amsterdam, The Netherlands; 3Department of Clinical Epidemiology, Amsterdam UMC, Academic Medical Center, University of Amsterdam, 1105 AZ Amsterdam, The Netherlands; a.h.zwinderman@amc.uva.nl; 4Department of Medical Biochemistry, Amsterdam UMC, Academic Medical Center, Amsterdam Cardiovascular Sciences, University of Amsterdam, 1105 AZ Amsterdam, The Netherlands

**Keywords:** resveratrol, Marfan syndrome, aortic aneurysms

## Abstract

Marfan syndrome (MFS) patients are at risk for cardiovascular disease. In particular, for aortic aneurysm formation, which ultimately can result in a life-threatening aortic dissection or rupture. Over the years, research into a sufficient pharmacological treatment option against aortopathy has expanded, mostly due to the development of rodent disease models for aneurysm formation and dissections. Unfortunately, no optimal treatment strategy has yet been identified for MFS. The biologically-potent polyphenol resveratrol (RES), that occurs in nuts, plants, and the skin of grapes, was shown to have a positive effect on aortic repair in various rodent aneurysm models. RES demonstrated to affect aortic integrity and aortic dilatation. The beneficial processes relevant for MFS included the improvement of endothelial dysfunction, extracellular matrix degradation, and smooth muscle cell death. For the wide range of beneficial effects on these mechanisms, evidence was found for the following involved pathways; alleviating oxidative stress (change in eNOS/iNOS balance and decrease in NOX4), reducing protease activity to preserve the extracellular matrix (decrease in MMP2), and improving smooth muscle cell survival affecting aortic aging (changing the miR21/miR29 balance). Besides aortic features, MFS patients may also suffer from manifestations concerning the heart, such as mitral valve prolapse and left ventricular impairment, where evidence from rodent models shows that RES may aid in promoting cardiomyocyte survival directly (SIRT1 activation) or by reducing oxidative stress (increasing superoxide dismutase) and increasing autophagy (AMPK activation). This overview discusses recent RES studies in animal models of aortic aneurysm formation and heart failure, where different advantageous effects have been reported that may collectively improve the aortic and cardiac pathology in patients with MFS. Therefore, a clinical study with RES in MFS patients seems justified, to validate RES effectiveness, and to judge its suitability as potential new treatment strategy.

## 1. Introduction

Marfan syndrome (MFS) is an autosomal dominant inherited disorder of the connective tissue, characterized by mutations in the Fibrillin-1 gene (*FBN1*). Patients with MFS predominantly suffer from skeletal, ocular and cardiovascular disease. The cardiovascular complications, in particular aortic aneurysm formation, ultimately result in aortic dissection or rupture, leading to reduced life expectancy [1]. For a rare disease, MFS is relatively common with an estimated prevalence of 1:5000 [2].

Standard management of the cardiovascular disease in MFS is to surgically resect the enlarged part of the aorta, which is then replaced by a synthetic graft (with or without an artificial aortic valve), when the aneurysm has reached certain dimensions (4.5–5 cm aortic diameter). Although this strategy has increased survival in MFS significantly, this surgery which is on average performed at a relatively young age (20–50 years), is a heavy burden. Moreover, prophylactic surgery of the proximal aorta is associated with progression of aneurysm formation and aortic dissection in the more distal aorta [3].

It is intriguing that in this era of drug development, no pharmacological treatment strategy has been identified that can inhibit aortic disease in MFS patients. Thus far, pharmacological treatment is based on blood pressure lowering drugs, using mostly β-blockers and the angiotensin II type-1 receptor (AGTR1) blocker, losartan. While these drugs slow down the aortic disease somewhat in MFS patients [4], evidence for the efficacy of these drugs on aortic root dilatation in patients is limited, as well as the evidence for these drugs to target the underlying cause of the progressive aortic degradation.

MFS research has expanded over the last 15 years because of development of MFS mouse models, however, this has not yet translated into novel treatment approaches to compensate for the *FBN1* gene defect that causes cardiovascular disease [5]. Part of the problem is that MFS is complicated to study because already 1847 different *FBN1* mutations have been reported, according to the Universal Mutation Database (last update 2014), causing heterogeneity in MFS phenotype [6]. Moreover, even within MFS families with the same mutation there is a large variation in disease pathogenesis [7], probably caused by genetic modifiers, such as common or rare genetic variants (polymorphisms). So to reach effectiveness in all different MFS patients, it is essential to find common ground as therapeutic approach. In that light, we found that the biologically-potent polyphenol resveratrol (RES) promoted aortic repair in one of the MFS mouse models (Fbn1C1039G/+ mice) [8]. While cardiovascular disease in MFS may be caused by different *FBN1* defects and modifying factors, promoting cardiovascular repair would benefit all types of MFS patients.

RES is a dietary supplement found in certain nuts and plants, best known in the skins of grapes. RES is usually produced by plants in response to stressors, such as pathogens. A wide range of beneficial effects has been shown in rodent models of disease, such as reducing oxidative stress, improved calcium handling and inhibition of pathological hypertrophic signaling [9]. We have shown in the Fbn1C1039G/+ MFS mice that RES is also effective at inhibiting the aortic root dilatation rate by affecting a mechanism different from AGTR1 or transforming growth factor beta (TGF-β) signaling, which is prominent in MFS [8].

The present overview will discuss recent RES studies in animal models of aortic aneurysm formation and heart failure, to investigate the different effects reported for RES on cardiovascular pathology relevant for MFS patients, to explore if a clinical study with RES in MFS patients is supportable.

## 2. Aorta

Among all clinical complications in MFS patients, aortic complications are the leading cause of morbidity and mortality. The aorta, which is the largest artery in the human body, supplies all vital organs with oxygenated blood containing nutrients. At the site of the aortic root, the heart pumps blood into the aorta under high pressure. Ordinarily, the aortic wall can withstand these pressures due to the well-organized structural extracellular matrix (ECM) protein network it contains. However, in patients with MFS, genetic defects in these structural proteins, mostly *FBN1* mutations, cause fundamental changes in this network of ECM proteins, rendering the patients vulnerable for aortic disease.

Various studies have been conducted to analyze the effect of RES on different cardiovascular diseases [9,10]. We will here discuss primarily the studies related to the effect of RES on aortopathy in relation to what is found in MFS patients. Interestingly, RES was beneficial against aortopathy in four different aortic aneurysm models; namely in the local periaortic application of calcium chloride (CaCl_2_)-model, in the local inter-aortic elastase infusion model, in the systemic chronic infusion of angiotensin-II (AngII) model and in the genetic *FBN1*-mutation (Fbn1C1039G/+) model of MFS [8,11,12,13]. The effect of RES on aneurysm development is examined in these studies, and the effect on aortic features relevant for MFS will be validated by literature and summarized.

### 2.1. Aortic Aneurysm

The aortic wall consists of three well-defined layers; the internal layer (intima), the middle layer (media) and the outer layer (adventitia) (Figure 1). The intima mainly consists of an endothelial cell layer (and a small number of smooth muscle cell (SMC) layers in the adult human aorta). The endothelial cells form the tight barrier between the blood and the vessel wall, and provide the cues for SMCs, dependent on shear stress, mechanical stress and circulating molecules that they sense. Between the intima and the media there is the internal elastic lamina that separates, and at the same time connects, the two layers.

The media occupies around 80% of the entire vessel wall surface and consists of concentric fenestrated lamellae of elastic fibers, together with SMCs embedded in a fine network of ECM, such as collagen fibrils. The media is mainly responsible for the contractility and distensibility (elasticity) of the aorta.

The adventitia is the external stent of the aorta, offering a strong tensile support with thick layers of collagen fibers maintained by fibroblasts, and it contains the vasa vasorum. From the adventitia, the vasa vasorum protrudes into the outer layers of the media to sufficiently provide the necessary oxygen and nutrients to the aortic SMCs. Via the external elastic lamina the adventitia is connected to the media. While fibrilline-1 is mostly known for its presence in elastic fibers in the media, it is also abundantly present and associated with collagen fibers in the adventitia [14].

The normal aortic diameter decreases when moving away from the aortic valve. In the ascending aorta the normal diameter is <2.1 cm and in the descending aorta <1.6 cm. However, in 60–80% of the adult MFS patients, the aortic root is dilated beyond 1.5 times the normal value as observed by echocardiographic imaging [15,16]. Aortic dilatation in MFS patients is mainly seen in the aortic root (sinus of Valsalva) and the average annual growth of the aortic root is approximately 0.35 mm/year [17]. Excessive aortic dilatation in MFS is triggered by failure of the ECM network in the aortic vessel wall to sustain physiological hemodynamic stress [18]. The aortic damage activates a multitude of different signaling pathways, of which many end in chronic activation of the extracellular signal-regulated kinases ERK1/2 in the murine MFS models [19,20,21].

RES has reduced aneurysm formation in four experimental animal models [8,11,12,13]. We have shown the effectiveness of RES on aortopathy in a MFS mouse model (Fbn1C10393G/+) despite continuous ERK1/2 signaling. MFS mice were treated with either RES (0.1 mg/mL), the AGTR1-blocker losartan (0.6 mg/mL) or placebo, for two months starting at two months of age when aortic disease was already present. Treatments were provided in the drinking water, by which the mice would consume approximately 0.2 mg RES/day. Indeed, when RES is administered to Fbn1C10393G/+ MFS mice, aortic dilatation is inhibited just as effectively as by losartan. However, where losartan reduced AGTR1-mediated activation, RES did not, but induced NAD-dependent deacetylase sirtuin-1 (SIRT1) activation as expected. SIRT1 is involved in cellular metabolism, such as enhancing the energetic status of the cell. Yet, when providing another SIRT1 activator, aneurysm formation could not be rescued, rendering room for discussion what the working mechanism of RES on reducing aortopathy may be.

The RES effect was associated with downregulation of detrimental aneurysm microRNA-29b (miR-29b) and improved elastin integrity and SMC survival [8]. MicroRNAs are small RNA molecules that bind specific mRNA gene transcripts, thereby targeting it for degradation and thus regulating gene expression. Since miR-29b plays a role in aortic aging [22], and reducing miR-29b could rescue abdominal aortic aneurysm formation [22,23] and MFS aortopathy [24] in mice, miR-29b may be central to combat aorta degeneration and promote SMC survival. In vitro we could mimic the RES effect on miR-29b synthesis by treating cultured endothelial cells with RES and providing their culture medium to SMCs, revealing the interplay between these two cell types [8]. Moreover, miR-21 was increased by RES in the aorta of MFS mice, which is in line with the anti-aneurysm effect of miR-21 observed in the AngII-induced aortic aneurysm model [25]. These miR’s have not been studied in the other three RES studies on aortic aneurysm formation.

All three rodent models where abdominal aortic aneurysms (AAA) were formed, showed that treatment with RES reduced AAA development or progression. AAA is characterized by chronic inflammation and vessel wall thinning due to severe ECM loss and SMC death. While the level of inflammation in MFS aortic tissue is minimal [26,27] compared to AAA, ECM degradation/remodeling and SMC function/viability plays a crucial role in AAA and MFS aneurysms [5].

Specifically, in the murine CaCl_2_-induced AAA model, 100 mg/kg RES or saline was given by intraperitoneal injection every day for six weeks (approximately 2.5 mg RES/mouse/day) [11]. The main mechanism for reduced AAA by RES was inhibition of inflammation, oxidative stress, angiogenesis, and elastic lamina degradation by a reduction in proteolytic activity of matrix metalloproteinase (MMP)2 and MMP9 [11].

In the elastase-induced rat AAA model, a similar phenotype was observed after RES administration [12]. Male Sprague Dawley rats were already treated with RES one week prior to induction of AAA, which was provided as 10 mg/kg/day RES continuously in their drinking water for three weeks (approximately 3 mg/rat/day). The main mechanism for AAA prevention by RES in this model was by counteracting the inflammatory response coinciding with a decrease in angiogenesis marker VEGF and protease MMP9 [12].

In the murine AngII-induced AAA model, the effect of RES on pre-established AAA (thus AAA progression) was studied [13]. The 14-week old male Apolipoprotein E deficient (hyperlipidemic) mice received continues AngII (1.0 µg/kg/min) infusion via osmotic minipumps for 8 weeks. AAA developed for 2 weeks, whereafter the mice were divided into treatment groups. One group received RES supplemented high fat diet (0.05 g RES/100 g diet, thus approximately 1 mg RES/mouse/day) and the other group high fat diet alone. RES treatment reduced AAA progression with induced SIRT1 activity, as observed in the MFS mice as well. Proinflammatory pathways were reduced upon RES, as well as MMP2 and -9. The authors were interested in the protective role of angiotensin-converting enzyme 2 (ACE2), which was increased in the mice and in cultured SMC by RES in a SIRT1-dependent manner.

Taken together, RES treatment showed multiple different mechanisms by which the aorta integrity and aortic dilatation was preserved. Below a more detailed discussion will be provided on the potential mechanisms of RES related to endothelial and SMC function in MFS.

### 2.2. Endothelial Dysfunction

RES reduced inflammation, angiogenesis and miR-29b in the above mentioned aneurysm models, which all related to diminishing endothelial cell activation. Interestingly, it has been shown that MFS patients have disturbed endothelial function, as measured by flow mediated dilatation, which strongly correlated with enhanced aortic diameter [28], posing the question if endothelial dysfunction is causally related to aneurysm formation.

#### 2.2.1. eNOS/iNOS Balance

The vascular endothelium determines the permeability of the aorta for macromolecules and leukocytes, prevents coagulation, and regulates vascular tone. The endothelium regulates vascular tone by communicating with SMCs via endothelial cell-derived nitric oxide (NO), which reduces SMC contractility [29]. Endothelial dysfunction is described as an impaired endothelium-dependent vasorelaxation caused by the loss or overproduction of NO bioavailability [30]. RES has been shown to regulate intracellular calcium in endothelial and SMCs differently. In endothelial cells it triggers NO synthesis [31]. NO is a potent vasodilator synthesized by endothelial NO synthase (eNOS), which is impaired in a MFS mouse model [29]. However, there are three isoforms of NOS; the endothelial, the neuronal and the inducible forms [30]. In contrast to eNOS production in endothelial cells, inducible NOS (iNOS) is increased in MFS in the SMC [32]. Excessive iNOS driven NO production causes oxidative stress and cellular damage via accumulation of peroxynitrites (ONOO-) [33]. The aortic iNOS and NO levels in the Fbn1C1039G/+ mouse model of MFS could be neutralized, which protected mice from aortic dilatation and medial degeneration [34]. The relevance of iNOS as driver of aneurysm formation also becomes clear in a novel MFS-like model with metalloproteinase ADAMTS1-deficiency, being essential to restrain iNOS activity [34]. Interestingly, in the MFS aorta ADAMTS1 was almost absent, which may explain the iNOS phenotype in the MFS mice. The effect of RES on ADAMTS proteins has not been investigated yet in the context of vascular disease.

The transcription factor KLF2 is essential in endothelial cell health by promoting a quiescent flow sensitive phenotype, in part by increasing eNOS [35]. Moreover, induction of KLF2 can reduce TGF-β–mediated activation of endothelium [36]. Since TGF-β is enhanced in MFS [37], this may be the cause of endothelial dysfunction in MFS. We and others demonstrated that RES enhances KLF2 [8] via SIRT1 activation [38], and multiple studies showed that RES increased eNOS [39,40,41], thereby promoting healthy endothelial cell function. Interestingly, miR-21 overexpression in endothelial cells also induced eNOS [42], so the induction of miR-21 upon RES in the MFS mice [8] probably played a role in endothelial de-activation as well.

#### 2.2.2. NOX4 and ROS

Apart from the reactive nitrogen species, there is also enhanced oxidative stress in MFS via other mechanisms [32,43,44]. Oxidative stress in the vessel wall may also be caused by the nicotinamide adenine dinucleotide phosphate (NADPH) oxidase, which is a membrane-bound enzyme complex consisting of different NOX subunits. This enzyme complex can become activated in all vascular cell types, endothelial cells, SMCs, and fibroblasts, upon inflammatory conditions [45]. Careful regulation of NADPH oxidase activity is crucial to maintain a healthy level of reactive oxygen species (ROS) in the vasculature. Since inflammatory cells have enhanced NADPH oxidase activity, causing excessive ROS, the protective effect of RES in the CaCl_2_-induced AAA model is presumably in part caused by inhibition of NADPH oxidase-mediated ROS from inflammatory cells and inflammation-induced activation of vascular cells. However, there is also accumulation of oxidative stress observed in MFS mice, which have limited inflammation. The excessive oxidative stress caused vasomotor dysfunction in the thoracic aorta [43]. RES is known to reduce oxidative stress, which is mostly ascribed to activation of SIRT1, causing inhibition of NADPH oxidase activation and thereby protecting endothelial function [46]. Inhibition of SIRT1 significantly increased vascular superoxide production, enhanced NADPH oxidase activity, and mRNA expression of its subunits p22(phox) and NOX4, which was prevented by RES [46]. Interestingly, NOX4 is highly increased in the human and murine MFS aorta. Moreover, in MFS mice deficient for NOX4, aortic aneurysm formation was attenuated and elastin degradation reduced [47]. Apart from beneficial SMC effects, endothelial dysfunction in the MFS ascending aorta was prevented by NOX4-deficiency or ROS inhibition [47].

While SIRT1 activation seems key in these pathways, we experimented with SIRT1 activation and SIRT1 inhibition in the Fbn1C1039G/+ MFS model and did not observe a beneficial or detrimental effect on the aorta; aortic dilatation was still observed [8], suggesting that perhaps multiple signaling pathways induced by RES act in concert to exert its beneficial function. In that light, we also described an endothelial-dependent effect of RES on the downregulation of detrimental miR-29b in SMC, thereby increasing SMC survival [8].

### 2.3. Medial Degeneration

The strict architecture of the medial aorta normally consists of well-organized elastic fibers, surrounded by circumferentially oriented SMCs and collagen fibers embedded in a complex ECM [48]. The main functions of the aortic media are contractility and distensibility. Mutations in SMC contractile proteins and ECM(-related) proteins are known to cause aortic aneurysm disease, showing the importance of both SMC contractility and ECM structure for aortic health [49].

Medial degeneration or cystic medial necrosis is a process observed in the aging aorta [50], patient with hypertension [50], bicuspid aortic valves [51] and Fallot’s tetralogy [52], but also specifically encountered in patients with aortic aneurysms, such as in MFS patients [53]. Medial degeneration in MFS is characterized by accumulation of proteoglycans and glucosaminoglycans within SMC-depleted areas of the aortic media, fragmentation or depletion of elastic fibers and SMC death [54]. The effect of RES on these features will be extracted below.

#### 2.3.1. ECM Degeneration

In all the aneurysm studies using RES, the ECM was protected or restored. Interestingly, miR-29b downregulation in different aneurysm models also resulted in improved ECM structure and prevented aneurysm growth [22,23,24]. We showed that RES decreased miR-29b in the MFS aorta in mice and promoted SMC survival [8]. However, others showed that miR-29b directly regulated ECM remodeling because many ECM genes are targets of miR-29b [22]. In addition, in the MFS mice, RES decreased elastin breaks while losartan treatment did not [8], which is in line with a RES-mediated miR-29b function in ECM protection. Thus increasing ECM production and remodeling by RES may overcome excessive ECM damage and thereby limit aneurysm growth.

Apart from ECM production, preventing ECM destruction by proteases may be a feasible strategy to reduce aortic disease. In AAA, the aortic ECM is destroyed by the abundance of aortic inflammation, where inflammatory cells secrete proteolytic enzymes to invade the vessel wall and to clear the surroundings from ECM debris. A well-studied class of proteases is the MMPs, which are abundant in human aneurysm tissue. MMPs form a family of enzymes that degrade ECM components and are released from mesenchymal cells and leukocytes [55]. High levels of MMPs result in a decreased stability of the ECM, and eventually in reduction of elasticity and rupture of the vessel wall. In all three AAA models using RES, MMP9 was reduced, which is mostly related to reduced macrophage influx, because these cells produce excessive MMP9 [11,12,13]. Since inflammation is very mild in MFS aortic tissue and inhibition of inflammation did not reduce aortic dilatation in MFS mice [26,27], most MMPs in MFS will be produced by the SMCs. Interestingly, in two of the AAA models using RES [11,13], also MMP2 was reduced upon RES treatment. Indeed, in a severe murine model of MFS, the Fbn1mgR/mgR mice, enhanced MMP2 production was observed in aortic SMCs, which was responsible for the detrimental induction of ERK1/2 and aortic dilatation [56]. The same authors generated MMP2 deficient MFS mice, which had prolonged lifespan due to decreased aortic rupture. Thus MMP2 inhibition seems a favorable feature of RES in MFS.

In the aneurysm research field doxycycline, an antibiotic, is often used as golden standard to reduce aneurysm formation in mice, which is ascribed to MMP inhibition. Yet, doxycycline actually inhibits mitochondrial function of all mammalian cells [57], since mitochondria are of bacterial origin. Thus apart from preventing MMP production, doxycycline changes cellular metabolism, which makes MMP inhibition not a specific function of doxycycline. Since mitochondrial dysfunction is characteristic for aneurysm tissue and causes oxidative stress by enhanced ROS [58], further suppression of mitochondrial respiration by long-term doxycycline treatment in humans does not seem constructive. Indeed, a human AAA trial using doxycycline was terminated prematurely due to enhance AAA growth in the doxycycline treated patients [59]. The mitochondrial dysfunction in the aneurysm tissue could however be reversed by promoting peroxisome proliferator-activated receptor gamma coactivator 1-alpha (PGC1a)-mediated mitochondrial biogenesis [58], which is also a well-known target and function of RES [60].

#### 2.3.2. SMC Death

SMCs are highly specialized cells that regulate blood pressure and blood flow through contraction and regulation of ECM structure [61]. Aortic SMCs exhibit a differentiated phenotype characterized by the expression of contractile markers specific to smooth muscle [62]. A unique and essential quality of the SMC is its ability to move between a synthetic and contractile phenotype. This is not only critical for normal development but also for response to injury through proliferation, migration and ECM synthesis. The relevance of proper SMC function is also exemplified by mutations in a number of SMC contractile genes and genes in the TGF-β pathway, causing aneurysm formation through impaired SMC response [49]. Moreover, the study where calcium channel blockers display increased aneurysm development in MFS patients and mice is also related to SMC contractility [20]. As has been discussed already, there is enhanced oxidative stress in the MFS aorta. Adrenergic contraction experiments with MFS aortas revealed that impaired vasomotor function was caused by oxidative stress due to enhanced iNOS and NADPH-oxidases and reduced superoxide-eliminating enzymes superoxide dismutase-1 and -2 [43]. Interestingly, SMC dysfunction precedes the elastin degradation [63], and increased SMC loss leads to aortic wall weakness in MFS [51]. This suggests that improving SMC health may increase aortic repair. In fact, in all four aneurysm manuscripts with RES treatment, the aortic elastin integrity was preserved, indicative of better SMC function, since these cells maintain the ECM in the aortic media.

There have been various observations concerning MFS SMC. They have been described as less differentiated leading to an immature vessel wall [64] or a premature switch from a synthetic to a contractile phenotype [65]. SMCs cultured from MFS aortas actually have an extremely differentiated SMC phenotype because of a TGF-β-induced SMC contractile gene expression profile [66]. This reveals that there are so many more SMC phenotypes apart from the two mostly described as contractile or synthetic. The MFS SMCs are stiff cells and impaired in regenerative capacity [66]. Thus it seems that the flexibility to switch phenotype is lost, so they are not responsive to signals to promote aortic repair. In line with these data, we found that the SMCs in the dilated aorta in MFS mice were senescent [8]. Senescence is characterized as a state of premature termination of the proliferative response often induced by oxidative stress. The presence of enhanced oxidative stress in MFS aortas has been discussed earlier. RES is known to counteract senescence and indeed changed SMC phenotype in MFS mice by increasing miR-21, which is known to stimulate SMC proliferation [25,67]. One of the targets of miR-21 is the phosphatase and tensin homolog (PTEN) protein expression. Increased miR-21 results in decreased PTEN, leading to increased activation of AKT, a cell survival factor.

Obviously, next to enhanced proliferation, decreased SMC death is important to prevent SMC depletion in the aorta. Interestingly, we previously observed that anti-inflammatory treatments in MFS mice did not reduce aortic dilatation. Actually, prednisolone treatment induced glucosaminoglycan accumulation in the aorta [27]. Since this is characteristic at sites of SMC depletion [54] it suggests that prednisolone promoted SMC death. In the RES treated MFS mice we showed increased pro-inflammatory nuclear factor kappa-light-chain-enhancer of activated B cells (NFĸB) signaling, which was probably responsible for the reduction in detrimental miR-29b, and therefore decreased SMC apoptosis [8,24]. The specific anti-apoptotic target transcripts of miR-29b were Bcl-2 and Mcl-1, which were increased in the RES treated MFS aorta, while SMC apoptosis was diminished. So where excessive inflammation is harmful in the AAA models, degrading the aorta faster than its repair capacity, modest inflammation may activate the SMC to enhance their repair function in MFS, overcoming the senescent state by promoting SMC proliferation and preventing death. RES reduced inflammation in the AAA models and decreased SMC death in MFS, showing again that RES influences multiple mechanisms.

In conclusion, RES improved endothelial dysfunction, ECM degradation and SMC death by alleviating oxidative stress (eNOS/iNOS balance; NOX4 and ROS), MMP2 and the miR-29b/miR-21 balance, respectively (Figure 1).

## 3. Heart

Apart from aorta pathology, other established features in MFS patients concern the heart; namely mitral valve prolapse (MVP), mild left ventricular (LV) dilatation (i.e., increased LV end diastolic dimension) and/or mild impairment of both LV systolic and diastolic function. The latter two can even occur in patients without significant valvular abnormalities or previous aortic surgery [68]. In addition, pulmonary artery dilatation may occur, often presenting simultaneously with dilatation of the aortic root [69]. The latter feature however, seldom results in symptoms in the patient. Due to a lack of specificity and incomplete knowledge of threshold values for dilatation of the pulmonary artery, this minor criterion was omitted from the revised criteria (Ghent 2) [70]. For this reason, we will only discuss MVP and LV dysfunction.

### 3.1. Mitral Valve Prolapse

Mitral dysfunction is present in 80% of patients with MFS, identified with auscultatory or echocardiographic evidence [71]. In contrast to idiopathic mitral valve regurgitation, the onset in MFS patients occurs at extremely young age and increases steadily with age [71,72]. MVP increases from 43% at 30 years of age to 77% at 60 years of age [72]. MVP is characterized by the systolic displacement or billowing of the mitral leaflets into the left atrium. MVP usually causes minor clinical symptoms in patients, but is a known risk factor for heart failure, arrhythmia, endocarditis, and sudden death [73]. Moreover, it is the most common cause of isolated mitral regurgitation requiring surgical repair.

While MVP is an important clinical feature in MFS patients, it has not been studied consistently in MFS rodent models. Moreover, there is no other model for MVP which included studies with RES. This is clearly a research opportunity. The only relevant data on mitral valves development in MFS was obtained from Fbn1-deficient MFS mice, where postnatal architectural changes in mitral valves were observed. The leaflets showed excess cell proliferation, reduced apoptosis, and increased TGF-β pathway activation [74]. The effect of RES on mitral valve architecture and function has not been studied yet.

### 3.2. Left Ventricular Impairment

Impairment of the left ventricular function in patients with MFS may occur as a consequence of significant valvular disease, but can also appear due to other severe vascular manifestations of MFS, like aortic dilatation. The latter can lead to increased hemodynamic stress and, combined with altered cardiac mechanobiology, result in cardiac dilatation [75]. However, in a small subset of MFS patients, impaired LV function, as expressed by increased LV diameters, has also been observed without a clear cause [76], suggesting an intrinsic cardiac defect due to the *FBN1* mutation. Indeed, it was demonstrated that a fibrillin-1-deficient ECM compromises the physical properties of myocardial tissue, resulting in abnormal mechanosignaling by muscle cells and subsequently leading to spontaneous dilated cardiomyopathy in MFS mice [77]. ECM defects thus make the heart vulnerable to cardiac dysfunction.

Fibrillin-1 is present as long fibers in the apex, mid-ventricles and atria. Collagen had a similar arrangement to that of fibrillin-1, whereas elastic fibers were primarily present in the atria and the blood vessels [78]. It is known that microfibrillins in the ECM are expressed in the left ventricle of the heart [78]. The Fbn1C1039G/+ mice demonstrated mild LV contractile dysfunction. Both structural ECM and molecular signaling alterations are implicated in MFS-related cardiomyopathy [79].

Since most of the rodent heart models where the effect of RES has been examined, are based on myocardial infarction or cardiac hypertrophy, the model representing dilated cardiomyopathy, occasionally observed in MFS, is the chemical toxicity model of doxorubicin (DOX)-induced thinning of the left ventricle. DOX is used as an antitumor drug, but its clinical application is limited because of its cardiotoxicity. The characteristics of this model are thinning and dilatation of the ventricular wall, causing reduced cardiac function presented with a reduced ejection fraction [80,81]. Therefore, we will use these DOX studies to extract the potential effects of RES on dilated cardiomyopathy known in MFS patients.

Several mechanisms have been proposed to explain the cardiotoxic side effects of DOX, including cardiomyocyte apoptosis, myofibrillar damage, impaired calcium handling, impaired mitochondrial activity, and increased generation of ROS [82]. As described previously, cell death, ECM damage, and increased oxidative stress were also observed in the aorta of MFS patients and mice, therefor it is likely that these pathological processes play an important role in the pathology of the heart as well.

In multiple rodent DOX-models, the effect of RES on cardiac function was determined, where RES protected the cardiomyocytes against structural damage and apoptosis [83,84,85,86]. This effect was mainly associated with the upregulation of SIRT1. In two studies, this upregulation led to reduced expression of tumor suppressor protein p53, which subsequently de-activated the apoptotic pathway [83,85]. Another model demonstrated that the SIRT1 upregulation due to RES, reduced mitogen-activated protein kinase p38 (p38MAPK) activation, which is the primary pathological signaling pathway in oxidative stress-induced heart injury, and in this way protected cardiomyocyte function [86]. Apart from alleviating DOX-induced cardiotoxicity through inhibiting p38MAPK/p53-mediated apoptosis, RES also improved autophagy by increasing AMP-activated protein kinase (AMPK) activation [87]. AMPK is an enzyme that plays a role in cellular metabolism when cellular energy is low. Autophagy is the process of intracellular vesicles engulfing protein aggregates or damaged mitochondria to fuse with lysosomes to degrade the contents and reuse the building blocks. This AMPK-mediated self-cleaning promotes cellular health [88]. This pathway may actually also apply to the aorta, since in endothelial cells, RES improved cellular health by promoting autophagy in an AMPK-dependent manner as well [89,90,91]. The effect of RES on AMPK-dependent autophagy in SMC remains to be determined.

DOX induces mitochondrial dysfunction and thereby increases oxidative stress in cardiomyocytes by enhanced ROS production, which also occurs in MFS, and is thought to cause cellular and ECM damage. RES was shown to decrease ROS production in the DOX-induced heart injury model [86]. The proposed mechanism for the decrease in ROS accumulation is the antioxidant properties of RES, by increasing the activity of the antioxidant enzyme superoxide dismutase, which converts ROS. The capacity of RES to reduce oxidative stress has similarly been observed in AAA [11].

In general, the data on the effect of RES that could be of interest for the heart features observed in MFS patients is limited. However, the main protective mechanisms of RES in the context of cardiac function concern SIRT1-mediated downregulation of p38MAPK activity to reduce cardiomyocyte apoptosis, AMPK-mediated enhanced autophagy and increased antioxidant superoxide dismutase to decrease oxidative stress/ROS (Figure 2).

## 4. Exercise Mimetic

Exercise is widely accepted to prevent or mitigate metabolic disease such as obesity, type 2 diabetes, and cardiovascular disease. For patients with MFS it is advised to refrain from doing intense sports to prevent a rise in blood pressure and thereby aortic aneurysm expansion and rupture, although hard evidence is lacking to support this advice. However, exercise mimetics have been identified, and may thus be of value as pharmacological agents. Interestingly, RES is considered an exercise mimetic [92,93], which puts exercise in the context of MFS in another light. Recently, two studies in MFS mice using an exercise protocol have shown to prevent aortic dilatation [94,95].

In the MFS Fbn1C1039G/+ mice at four months of age, there are already differences between the MFS mice and the wild type littermates in aorta dilatation, elastic lamina ruptures, aortic stiffness, as well as cardiac hypertrophy, left ventricular fibrosis, and intramyocardial vessel remodeling. A moderate exercise protocol reduced aorta dilatation and cardiomyopathy in the MFS mice, while the other pathological characteristics remained unaltered [94]. The exercise training consisted of running on a treadmill for 1 h a day, five days a week for five months. However, the study could not reveal the underlying mechanisms for the beneficial effects of exercise on the MFS aorta and heart.

In the same MFS mouse model, yet another research group performed exercise experiments with two different protocols; voluntary cage-wheel exercise or forced treadmill exercise, and a sedentary lifestyle as reference for aorta pathology, for five months starting at four weeks of age [95]. They demonstrated that exercise could prevent aorta dilatation, preserve the elastic fibers and improve the tensile strength of the aortic wall as compared to the sedentary MFS mice. With the voluntary exercise regimen, optimal aortic improvements were measured due to reduced MMP2 and MMP9 activity.

If the main beneficial effect of RES is related to the exercise effect, then other exercise mimetics may also be beneficial in MFS. The anti-diabetic drug metformin, and AMP analog 5-aminoimidazole-4-carboxamide ribonucleotide (AICAR), and nuclear receptor peroxisome proliferator-activated receptor delta (PPARδ) agonists have also been described as exercise mimetics [92,93], which may thus affect cardiovascular pathology in MFS similarly. The exercise signature, using these exercise mimetics, consists of enhanced mitochondrial biogenesis in which SIRT1, AMPK, and PGC1α play an important role [92,93], and were all mentioned earlier as protectors in MFS studies.

Interestingly, prescription for metformin is indeed associated with a decrease in AAA enlargement [96], already suggesting a beneficial role for metformin in aneurysm progression. This effect deserves more extensive follow up in prospective clinical trials in AAA patients and perhaps also in MFS.

Overall, the mechanisms during mild/moderate exercise or RES treatment that protect MFS mice from cardiovascular pathology may be the same and thus setting up an exercise training protocol for MFS patients could be considered. In the meantime, a clinical study using RES in MFS patients seems justified and feasible.

## 5. Conclusion

The cardiovascular pathology in MFS patients is complicated. Aortic pathology mainly consists of uncontrolled aortic growth, endothelial dysfunction, and medial degeneration. Besides aorta pathology, MFS patients may present with other cardiovascular manifestations, such as left ventricular dysfunction. Given the multifactorial nature of MFS, traditional therapies like β-blockers and losartan likely do not adequately target all pathways involved in cardiovascular dysfunction in MFS. In the literature we found evidence of beneficial effects of RES on multiple levels to combat aneurysm formation and left ventricular dysfunction in multiple rodent-based animal models, suggesting that RES may have therapeutic potential in MFS patients by reducing oxidative stress and promoting cell survival. Although, clinical trials with RES have been performed, these were not in MFS patients. While in rodents RES is very potent to combat a plethora of diseases, the translation to human cardiovascular disease seems less efficient [97]. Yet, the rapid implementation of RES in a well-designed trial, over a longer period of time than previously performed with RES, seems attractive to determine whether treatment with RES is beneficial for MFS patients.

With the variable results from different ARB trials in mind over the last decade, careful consideration on the setup of a new clinical trial in MFS patients is required. A meaningful strategy is largely determined by the number of patients available, the chosen primary outcome, the duration of the study and the dosage of RES. The collective data in this manuscript shows that a feasibility study could be considered to monitor specific RES effects in MFS patients as prerequisite for a novel clinical RES trial.

## Figures and Tables

**Figure 1 ijms-20-01122-f001:**
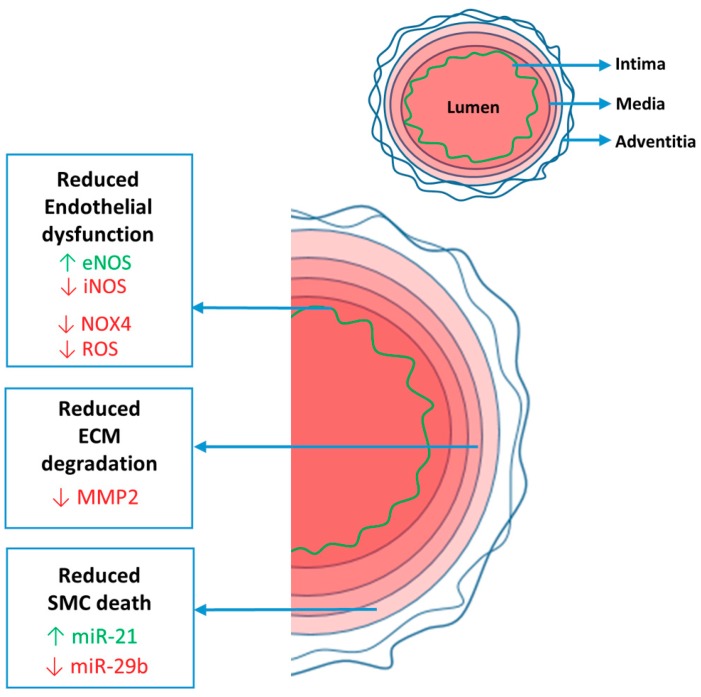
Resveratrol improves pathological aortic features in Marfan syndrome by enhancing beneficial (green) eNOS and miR-21 and decreasing detrimental (red) iNOS, NOX4, ROS, MMP2 and miR-29b. eNOS or iNOS: endothelial or inducible nitric oxide synthase; miR-21 and miR-29b: microRNA-21 and -29b; NOX4: NADPH oxidase 4; ROS: reactive oxygen species; MMP2: matrix metalloproteinase 2.

**Figure 2 ijms-20-01122-f002:**
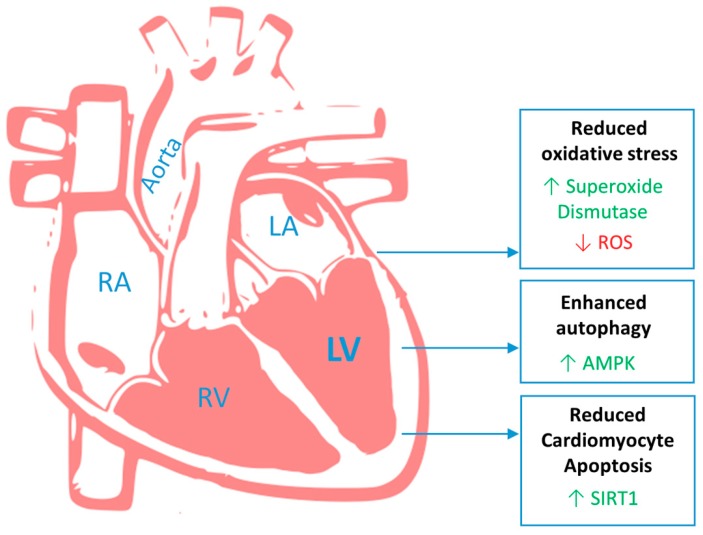
Resveratrol improves pathological cardiac features in Marfan syndrome by increasing beneficial (green) superoxide dismutase, AMPK and SIRT1 and decreasing detrimental (red) ROS. ROS: reactive oxygen species; AMPK: AMP-activated protein kinase; SIRT1: sirtuin-1; RA: right atrium; RV right ventricle; LA: left atrium; LV: left ventricle.

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
