# Peer review of "The Potential Beneficial Effects of Resveratrol on Cardiovascular Complications in Marfan Syndrome Patients–Insights from Rodent-Based Animal Studies"

_ijms, 2019, doi:10.3390/ijms20051122_

Reviewer 1 Report

Title:

I would suggest to change the title to avoid misdirection of the readers: The authors discuss potential beneficial effects of Resveratrol in rodent, which could possibly lead to a study in Marfan patients. That’s why the title should be more like:

Are there positive effects of  Resveratrol on the cardiovascular system of Marfan patients?- insights from rodent-based animal studies

That’s what the authors are discussing in their abstract and the rest of the manuscript.

Abstract:

20: With regard to the guidelines, there is a strictly recommended medical treatment of MFS: beta blockers and AGTR1 blocker. As the authors want to discuss RES as new drug for the cardio-vascular impact of MFS, they need to compare A and B, that’s how drug studies are performed, right? Please clarify.

23ff: All those effects and mechanism are kind of cellular common placed. Please focus on the most relevant studies and report these results here.

28 ff: The same as above; up to now, I see a lack specificity regarding the potentially beneficial effects of RES.

Introduction:

48 ff: If the ascending aorta isn’t involved, a thoracolaparotomy is required for TAAA repair. Please change.

 52ff: It is not completely right that the intake of BB and AGTR1B doesn’t influence the underlying cardiovascular pathomechanism. You can only state this if you can prove it by good references- up to now, there is no reference in your manuscript.

 Aorta:

81: please rephrase.

82: Al à all

Figure 1: the subtitle occurs twice. Please correct.

325: aortas à aorta

 Heart:

3.1: MVP is of extreme clinical relevance for all MFS patients, correct. Yet, as you cannot discuss any studies focusing on MVP in a rodent model and RES, why don’t skip it?

Figure 2: The quality of the figure is poor in my version. Can it be/ does it need a better resolution?

I really enjoyed reading your review. I would recommend to summarize all those facts and to present a multiple-study concept of the RES in MFS patients-study you mentioned in your abstract.

 Author Response

We thank all reviewers and the editors for their efforts, the careful appraisal of our manuscript, and the useful comments. We have replied the queries point-by-point (in a word document). Changes in the manuscript are visible through track changes.

REVIEWER 1

Title
I would suggest to change the title to avoid misdirection of the readers: The authors discuss potential beneficial effects of Resveratrol in rodent, which could possibly lead to a study in Marfan patients. That’s why the title should be more like:

Are there positive effects of  Resveratrol on the cardiovascular system of Marfan patients?- insights from rodent-based animal studies

That’s what the authors are discussing in their abstract and the rest of the manuscript.
Thank you very much for this relevant point and the given suggestion. In response to this we changed the title to:

‘The potential beneficial effects of Resveratrol on cardiovascular complications in Marfan Syndrome patients – insights from rodent-based animal studies.’  

Abstract:
20: With regard to the guidelines, there is a strictly recommended medical treatment of MFS: beta blockers and AGTR1 blocker. As the authors want to discuss RES as new drug for the cardio-vascular impact of MFS, they need to compare A and B, that’s how drug studies are performed, right? Please clarify.
Thank you for identifying this issue. Although it would be interesting to compare Resveratrol to either β-blockers or AGTR1-blockers in MFS patients, our Ethical Committee does not allow us to provide either β-blockers or AGTR1-blockers versus Resveratrol. That would be considered unethical in MFS patients. It will be possible, however, to propose a trial where Resveratrol will be used on top of the ongoing medical treatment as we previously did for losartan in the COMPARE trial. 

We now discuss clinical options briefly at the end of the manuscript:

“With the variable results from the different ARB trials in mind over the last decade, careful consideration on the setup of a new clinical trial in MFS patients is required.  A meaningful strategy is largely determined by the number of patients available, the chosen primary outcome, the duration of the study and the dosage of RES. The collective data in this manuscript shows that a feasibility study could be considered to monitor specific RES effects in MFS patients as prerequisite for a novel clinical RES trial.”

23ff: All those effects and mechanism are kind of cellular common placed. Please focus on the most relevant studies and report these results here.

28 ff: The same as above; up to now, I see a lack specificity regarding the potentially beneficial effects of RES.
We made the following changes in the abstract:

Abstract: Marfan syndrome (MFS) patients are at risk for cardiovascular disease. In particular for aortic aneurysm formation, which ultimately can result in a life-threatening aortic dissection or rupture. Over the years, research into a sufficient pharmacological treatment option against aortopathy has expanded, mostly due to the development of rodent disease models for aneurysm formation and dissections. Unfortunately no optimal treatment strategy has yet been identified for MFS. The biologically-potent polyphenol Resveratrol (RES), that occurs in nuts, plants and the skin of grapes, was shown to have a positive effect on aortic repair in various rodent aneurysm models. RES demonstrated to affect aortic integrity and aortic dilatation. The beneficial processes relevant for MFS included the improvement of endothelial dysfunction, extracellular matrix degradation and smooth muscle cell death. For the wide range of beneficial effects on these mechanisms, evidence was found for the following involved pathways; alleviating oxidative stress (change in eNOS/iNOS balance and decrease in NOX4), reducing protease activity to preserve the extracellular matrix (decrease in MMP2), and improving smooth muscle cell survival affecting aortic ageing (changing the miR21/miR29 balance). Beside aortic features, MFS patients may also suffer from manifestations concerning the heart, such as mitral valve prolapse and left ventricular impairment, where evidence from rodent models show that RES may aid in promoting cardiomyocyte survival directly (SIRT1 activation) or by reducing oxidative stress (increasing superoxide dismutase) and increasing autophagy (AMPK activation).

This overview discusses recent RES studies in animal models of aortic aneurysm formation and heart failure, where different advantageous effects have been reported that may collectively improve the aortic and cardiac pathology in patients with MFS. Therefore, a clinical study with RES in MFS patients seems justified, to validate RES effectiveness, and to judge its suitability as potential new treatment strategy.

 Introduction:
48 ff: If the ascending aorta isn’t involved, a thoracolaparotomy is required for TAAA repair. Please change.
We rephrased the sentence into the following (67):

‘’Although this strategy has increased survival in MFS significantly, this type of surgery which is on average performed at a relatively young age (20-50 years), is a heavy burden.’’

52ff: It is not completely right that the intake of BB and AGTR1B doesn’t influence the underlying cardiovascular pathomechanism. You can only state this if you can prove it by good references- up to now, there is no reference in your manuscript.
We have now revised the sentence to the following, to point out the limited evidence (75):

‘’ While these drugs slow down the aortic disease somewhat in MFS patients[4], evidence for the efficacy of these drugs on aortic root dilatation in patients is limited, as well as the evidence for these drugs to target the underlying cause of the progressive aortic degradation.’’

Aorta:
81: please rephrase.
We rephrased the following sentence "Among all clinical complications in MFS patients, aorta pathology is causing most lethality. " to (104):

‘’Among all clinical complications, aortic complications are the leading cause of morbidity and mortality in patients with MFS.‘’

82: Al à all
Thank you for noticing. We changed "al" to "all".

Figure 1: the subtitle occurs twice. Please correct.
We corrected the duplicate subtitle.

325: aortas à aorta
Thank you for noticing. We changed "aortas" to "aorta".

Heart:
3.1: MVP is of extreme clinical relevance for all MFS patients, correct. Yet, as you cannot discuss any studies focusing on MVP in a rodent model and RES, why don’t skip it?
We fully understand the Reviewers remark and have considered taking out the MVP section. However, then it seems that we overlooked MVP as pathological feature. Thus we have now chosen to emphasize the fact that MVP has not been studied in MFS rodent models consistently, let alone to study the effect of RES on MVP.

We have added the following sentence to the manuscript:

“While MVP is an important clinical feature in MFS patients, it has not been studied consistently in MFS rodent models. Moreover, there is no other model for MVP which included studies with RES. This is clearly a research opportunity. The only relevant data on mitral valves development in MFS was obtained from Fbn1-deficient MFS mice, where postnatal architectural changes in mitral valves were observed.”

Figure 2: The quality of the figure is poor in my version. Can it be/ does it need a better resolution?
We optimized the resolution to 600. 

I really enjoyed reading your review. I would recommend to summarize all those facts and to present a multiple-study concept of the RES in MFS patients-study you mentioned in your abstract.
While we are reluctant to already include an actual proposition of an RCT in the manuscript yet, we are pleased to let you know that we are currently working out the feasibility of a clinical study to investigate the potential beneficial effects of RES in MFS patients. A meaningful strategy is now debated and largely determined by the number of patients available, the duration of the study, the chosen primary outcome, and the dosage of RES. When we start a RES study in MFS patients, we will publish our chosen strategy. As mentioned previously, we have now added a brief section on our view for a clinical study at the end of the manuscript.

Reviewer 2 Report

This manuscript is a review on the role of natural herbal compound resveratrol (RES) in attenuation of vascular phenotype in Marfan syndrome (MFS). This manuscript is well written and structured, describing state of art gathered using animal models inducing aortopathy, especially a recent mice model of MFS with Fbn1 mutation. My comments to this manuscript are as follows:
1. The main message of this review is in my opinion far too enthusiastic. There is plethora of studies on RES at cellular and animal models. But intervention using RES diet supplementation did not bring any spectacular results in clinics (metaanalysis Fogacci et al. Rev Food Sci Nutrit 2018). This should be mentioned.

2. Induction of sirtuine-1 seeeds to be driving effects of RES, but in Authors own experiments this mechanism was rather dismissed. Could it be commented more about refernces 38, 46, 83, 86, 89 if alternate inducer of SIRT1 expression was used in control to RES.

3. The sentence in lines 291-292 starts with “The typical aneurism mutations...” suggest that ACTA2, MYH11, PRKG1 are typical findings in heritable aortic aneurism. I suggest rewriting this sentence, especially because Authors even do not mention Loeys-Dietz, a more common one. Also reference for this sentence [39] unnecessarily give the first author given name “Dianna”.

4. If the aim of the authors is to propose RCT on aortopathy prevention, this could be mentioned with an attempt to suggest daily RES dose and patients group size to achieve statistical power.

Minor comments:

Some minor English corrections:

l. 88 – casing most lethality (the)

l.118, 358, 418 – familiar with MFS (affected by?)

Author Response

We thank all reviewers and the editors for their efforts, the careful appraisal of our manuscript, and the useful comments. We have replied the queries point-by-point (in a word document). Changes in the manuscript are visible through track changes.

REVIEWER 2:

This manuscript is a review on the role of natural herbal compound resveratrol (RES) in attenuation of vascular phenotype in Marfan syndrome (MFS). This manuscript is well written and structured, describing state of art gathered using animal models inducing aortopathy, especially a recent mice model of MFS with Fbn1 mutation. My comments to this manuscript are as follows:

1. The main message of this review is in my opinion far too enthusiastic. There is plethora of studies on RES at cellular and animal models. But intervention using RES diet supplementation did not bring any spectacular results in clinics (meta-analysis Fogacci et al. Rev Food Sci Nutrit 2018). This should be mentioned.
Thank you for your concern about the main message. We agree that clinical trials were controversial. Nevertheless, they did show beneficial effects in some important pathways, especially when used in high dose. In general, well-designed trials are needed to determine the effect of Resveratrol. We revised the conclusion to your suggestion to the following:

 The cardiovascular pathology in MFS patients is complicated. Aortic pathology mainly consists of uncontrolled aortic growth, endothelial dysfunction and medial degeneration. Besides aorta pathology, MFS patients may present with other cardiovascular manifestations, such as left ventricular dysfunction. Given the multifactorial nature of MFS, traditional therapies like β-blockers and losartan likely do not adequately target all pathways involved in cardiovascular dysfunction in MFS. In the literature we found evidence of beneficial effects of RES on multiple levels to combat aneurysm formation and left ventricular dysfunction in multiple rodent-based animal models, suggesting that RES may have therapeutic potential in MFS patients by reducing oxidative stress and promoting cell survival. Although, clinical trials with RES have been performed, these were not in MFS patients. While in rodents RES is very potent to combat a plethora of diseases, the translation to human cardiovascular disease seems less efficient (Fogacci, F., et al., Effect of resveratrol on blood pressure: A systematic review and meta-analysis of randomized, controlled, clinical trials. Crit Rev Food Sci Nutr, 2018: p. 1-14) Yet, the rapid implementation of RES in a well-designed trial, over a longer period of time than previously performed with RES, seems attractive to determine whether treatment with RES is beneficial for MFS patients.  

 2. Induction of sirtuine-1 seeds to be driving effects of RES, but in Authors own experiments this mechanism was rather dismissed. Could it be commented more about references 38, 46, 83, 86, 89 if alternate inducer of SIRT1 expression was used in control to RES.
The Reviewer has correctly pointed out that we previously found no beneficial effect of SIRT1 activation (or detrimental effect of SIRT1 inhibition). Our view on this topic is that RES is metabolized, and in theory different metabolites may affect different mechanisms. But even if we consider only one metabolite as the active compound, it may still influence different pathways. While SIRT1 is the common readout of RES activity, we cannot rule out that other SIRT-independent pathways are also modulated by RES. Perhaps the accumulation of multiple beneficial events causes a beneficial effect on the aorta, which cannot be achieved by modulation of SIRT1 alone. More like a multifactorial disease, where just reducing one factor may not be sufficient to cure the patient.

3. The sentence in lines 291-292 starts with “The typical aneurism mutations...” suggest that ACTA2, MYH11, PRKG1 are typical findings in heritable aortic aneurism. I suggest rewriting this sentence, especially because Authors even do not mention Loeys-Dietz, a more common one. Also reference for this sentence [39] unnecessarily give the first author given name “Dianna”.
Thank you for this remark. We rephrased the sentence to:

‘’The relevance of proper SMC function is also exemplified by mutations in a number of SMC contractile genes and genes in the TGF-β pathway, causing aneurysm formation through impaired SMC response.’’

Thank you for indicating the citation error in reference [49].

4. If the aim of the authors is to propose RCT on aortopathy prevention, this could be mentioned with an attempt to suggest daily RES dose and patients group size to achieve statistical power.

While we are reluctant to already include an actual proposition of an RCT in the manuscript yet, we are pleased to let you know that we are currently working out the feasibility of a clinical study to investigate the potential beneficial effects of RES in MFS patients. A meaningful strategy is now debated and largely determined by the number of patients available, the duration of the study, the chosen primary outcome, and the dosage of RES. When we start a RES study in MFS patients, we will publish our chosen strategy.

We now discuss our view on clinical options briefly at the end of the manuscript:

“With the variable results from the different ARB trials in mind over the last decade, careful consideration on the setup of a new clinical trial in MFS patients is required.  A meaningful strategy is largely determined by the number of patients available, the chosen primary outcome, the duration of the study and the dosage of RES. The collective data in this manuscript shows that a feasibility study could be considered to monitor specific RES effects in MFS patients as prerequisite for a novel clinical RES trial.”

Minor comments:
Some minor English corrections:
l. 88 – casing most lethality (the)
We rephrased the following sentence "Among all clinical complications in MFS patients, aorta pathology is causing most lethality. " to:

‘’Among all clinical complications, aortic complications are the leading cause of morbidity and mortality in patients with MFS.‘’

l.118, 358, 418 – familiar with MFS (affected by?)
We changed ‘’familiar with’’ to
118 – adult MFS patients
358 – patients with MFS
418 – patients with MFS

Reviewer 3 Report

In this review entitled "The effect of Reservatrol on the cardiovascular system in relation to Marfan Syndrome" Mitzi M. van Andel et al. gave a detailed description about the research involved in the beneficial effects of RES in cardiovascular complications observed in animal models of Marfan syndrome. The review is well written with updated research in the field. 

Here are my minor comments.

Minor language corrections to be done.  Ex. In line 82, "al" was written instead of "all".      

I strongly suggest authors reconsider their title for this review. I suggest the title to be more specific like "cardiovascular complications" or "cardiovascular abnormalities" or "aortic abnormalities"   in Marfans syndrome rather than the general term like "cardiovascular system". I also suggest using word "protective effects of Resveratrol". These changes may make the title catchy for the readers.          

Author Response

We thank all reviewers and the editors for their efforts, the careful appraisal of our manuscript, and the useful comments. We have replied the queries point-by-point (in a word document). Changes in the manuscript are visible through track changes.

REVIEWER 3

In this review entitled "The effect of Resveratrol on the cardiovascular system in relation to Marfan Syndrome" Mitzi M. van Andel et al. gave a detailed description about the research involved in the beneficial effects of RES in cardiovascular complications observed in animal models of Marfan syndrome. The review is well written with updated research in the field. 

Here are my minor comments.

Minor language corrections to be done.  Ex. In line 82, "al" was written instead of "all".
Thank you for noticing. We changed "al" to "all".

I strongly suggest authors reconsider their title for this review. I suggest the title to be more specific like "cardiovascular complications" or "cardiovascular abnormalities" or "aortic abnormalities" in Marfans syndrome rather than the general term like "cardiovascular system". I also suggest using word "protective effects of Resveratrol". These changes may make the title catchy for the readers. 
Thank you for these suggestions. We now changed the title to:

‘’The potential beneficial effects of Resveratrol on cardiovascular complications in Marfan Syndrome patients – insights from rodent-based animal studies.’’